# The Lived Experience of Caregivers in the Older Stroke Survivors’ Care Pathway during the Transitional Home Program—A Qualitative Study

**DOI:** 10.3390/ijerph21101276

**Published:** 2024-09-25

**Authors:** Mayra Veronese, Silvio Simeone, Michele Virgolesi, Cristiana Rago, Ercole Vellone, Rosaria Alvaro, Gianluca Pucciarelli

**Affiliations:** 1Department of Biomedicine and Prevention, University of Rome Tor Vergata, 00133 Rome, Italy; veronese.mayra@gmail.com (M.V.); ragocristiana@gmail.com (C.R.); ercole.vellone@uniroma2.it (E.V.); rosaria.alvaro@gmail.com (R.A.); 2Departement of Clinical and Experimental Medicine, University of Magna Grecia Catanzaro, 88100 Catanzaro, Italy; silvio.simeone@unicz.it; 3Department of Public Health, University of Naples Federico II, 80131 Naples, Italy; michele.virgolesi@libero.it

**Keywords:** caregiver, transitional care, stroke, qualitative research

## Abstract

Background: stroke is a major cause of disability and death, globally. Many stroke survivors live with disabilities, requiring significant caregiving support. Caregivers often feel unprepared and burdened, experiencing isolation and health declines. Their well-being and involvement in discharge planning impact post-discharge care quality. Purpose: to investigate the experiences of caregivers of older stroke survivors regarding their expectations in the care pathway during the transitional home program, as this phenomenon is currently understudied in the present context. Methods: by employing a qualitative design, this study utilized a phenomenological approach developed by Cohen. Eighteen caregivers of stroke survivors participated. Findings: four main themes emerged from the analysis: (1) the sense of loss experienced by stroke survivors and perceived by caregivers; (2) the importance of social support in the recovery and well-being of both stroke survivors and caregivers; (3) the increased workload of caregivers; and (4) the lack of awareness of the needs during the transition, leading to resignation when facing complications. Conclusion: the findings highlight the significant impact of stroke on caregivers, affecting both the individual characteristics and social relationships of stroke survivors and their caregivers. There is a need for a nuanced understanding of caregivers’ roles and responsibilities.

## 1. Introduction

Stroke represents one of the most significant contributors to disability and mortality worldwide. Indeed, each year, more than 9.4 million individuals in the United States [1] and 1.12 million people in Europe [2] experience strokes, with the incidence increasing significantly with advanced age in both males and females [1]. In terms of incidence, approximately 795,000 individuals in the United States suffer from either recurrent or first-time strokes, and 76% of these are first attacks [1]. Nearly 60% of stroke survivors globally are living in the community with some level of disability [3].

A stroke has a significant impact on the health of survivors [4,5]. Depending on the location and type of stroke, patients may develop physical, psychological, or emotional disabilities [6], such as mood disorders, aphasia, dysphagia, and depression [3,7]. Approximately 80% of stroke survivors experience limitations in their daily life [3]. These disabilities also affect caregivers, who play a crucial role in the transition to home care, often feeling unprepared for the task. This lack of preparation, as reported in several studies [8,9], can become a significant burden on caregivers. Many of them describe feelings of isolation, abandonment, loneliness, and a gradual decline in their physical and mental health due to the increased workload. The management of post-discharge care is strongly influenced by the caregivers’ psychological and emotional well-being, resilience, and degree of involvement in discharge planning. Therefore, it is essential to implement tailored interventions to enhance caregivers’ preparedness during the transition to home care, aiming to achieve better outcomes for both stroke survivors and caregivers. A shift in rehabilitation philosophy, toward greater attention and support for both the patient and the caregiver, could lead to improved long-term outcomes. Moreover, the dyadic conceptual framework in stroke populations, developed by Savini et al. [10], identifies caregiver preparedness as a moderating factor that can improve the quality of life of both survivors and caregivers. Thus, there is a need to develop specific interventions that involve both stroke survivors and caregivers.

Several studies have analyzed the living experiences of caregivers during the discharge of stroke survivors [11,12,13,14]. For example, Gustafsson and Bootle [14] explored the shared experience of stroke caregivers during the transition to the home environment, shedding light on both fulfilled needs and encountered challenges following the discharge. Caregivers exhibit a great dedication to adjusting to new responsibilities and daily routines, and play an important role in rehabilitation. In a study by Chen et al. [11], the authors observed that caregivers showed profound uncertainty regarding their role and transition to care responsibilities post-hospital discharge, and the resilience of stroke survivors was augmented by their functional accomplishments and family support, whereas caregivers’ resilience was constructed upon their internal strength, past life experiences, and pre-existing coping strategies.

Although several qualitative studies [11,14,15] have been conducted on stroke caregivers during the transitional care period, many of these have focused on understanding the role of caregivers, their preparation with respect to managing the post-discharge, the uncertainties they face, and the factors contributing to their resilience. This represents a gap in the literature, because understanding the expectations of stroke caregivers during transitional home care could be fundamental for nurses and physicians to better develop a tailored intervention to improve caregivers’ transition from the hospital to the home. In addition, knowing the lived experiences of stroke caregivers during transitional care could be crucial for nurses and physicians because it allows for a more comprehensive approach to support both caregivers and stroke survivors throughout the recovery process, ultimately enhancing the effectiveness of post-discharge care interventions and promoting better outcomes for both parties.

The aim of this qualitative study is to delve into the experiences of caregivers of elderly stroke survivors, focusing specifically on their expectations and perceptions within the care pathway during the transitional home program. This investigation seeks to explore the emotional, psychological, and practical challenges faced by caregivers. By gaining a comprehensive understanding of these experiences, the study aims to provide evidence-based recommendations to enhance the support offered to caregivers during this critical transitional phase. The research question that guided the study was: ‘How do caregivers of older stroke survivors perceive and experience the transitional home care period, and what challenges do they face during this process?’

## 2. Materials and Methods

### 2.1. Study Design

A qualitative study was conducted using semi-structured, face-to-face interviews to explore the lived experiences of caregivers of older stroke survivors. The theoretical framework underlying this research was a phenomenological approach developed by Cohen [16]. This approach combines interpretive (Gadamerian) and phenomenological (Husserlian) approaches. Phenomenology is an inductive qualitative research rooted in a twentieth-century tradition [17], which is a form of inductive qualitative research. As proposed by Husserl, the founder of this methodology, phenomenology involves the suspension of all suppositions and is closely tied to consciousness, centering on the meaning derived from individual experiences. Descriptive phenomenology entails the detailed portrayal of daily experiences, with preconceived opinions intentionally set aside and bracketed by tradition [17]. In contrast, interpretative qualitative research is a qualitative approach based on examinations of personal lived experiences. This methodology has proved to be particularly valuable for investigating subjects that are intricate, ambiguous, and emotionally charged. This method, used in a prior study [18], was chosen because of its suitability in terms of gaining a deeper understanding of both lived experiences and the meaning attributed to such experiences by a family.

### 2.2. Study Settings and Recruitment

Convenience sampling was used with participants recruited in two Neurology wards at Azienda Ospedale-Università Padova, Italy, between May and July 2023. Eighteen participants were purposefully selected from the survey study and invited to partake in the descriptive qualitative study. Prior to signing the consent form, the caregivers were advised of the study’s aim and nature. They were guaranteed complete confidentiality in all phases of the study and assured that the reported data would contain no identifying information. Participants were guaranteed the opportunity to retire from the study at any time. Stroke caregivers were also advised that their refusal to participate in the study or withdrawal from it would not in any way undermine the quality of care provided.

### 2.3. Inclusion and/or Exclusion Criteria

Eligibility criteria for stroke caregivers included: (1) being 18 years of age or older, (2) being informal caregivers of stroke survivors—informal caregivers are typically family members who provide unpaid care to individuals with whom they have personal relationships [19], and (3) being involved during the stroke survivors’ transitional care from the hospital to the home.

### 2.4. Data Collection

Participants were interviewed once during the first month after discharge, a critical period in which caregivers must quickly adapt to providing care for the stroke survivor at home. The interviews were carried out in the participants’ native language (Italian), employing a video conferencing platform chosen individually by each participant based on their level of confidence. Following a comprehensive disclosure to the stroke caregivers regarding the study’s objectives and its nature, their explicit authorization was obtained for participation. Complete confidentiality was assured at all stages of the study, the collected data did not contain any identifying information. All participants had the option to withdraw from the study at any time. Pseudonyms were assigned to each participant at the time of the interview and then responses were encrypted before data analysis.

The selection and ordering of questions were shaped by the natural progression of the conversation and by the interviewee’s responses. This approach allowed for a more organic and adaptable interview process, ensuring that the questions remained relevant and tailored to the specific context and insights provided by each participant [20].

The interview utilized open-ended questions, allowing the interviewer to steer the conversation. In addition, the interviewers had no prior contact with the participants before this study. A semi-structured interview guide was developed, informed by existing literature and feedback from healthcare professionals experienced in qualitative research and stroke transitional care. The interview guide focused on themes related to caregivers’ experiences during the patient return home, the primary challenges encountered, the types of support required, and potential interventions needed prior to discharge to enhance preparation for home reintegration. Additionally, interviewees were encouraged to discuss any other significant aspects of the home transition period that had not been previously addressed. The session was concluded once participants had no additional input to provide. Each interview was audio-recorded and ranged in duration from 30 min to 40 min. Data saturation was achieved with 18 interviews after two independent researchers acknowledged that no new codes had emerged.

### 2.5. Data Analysis

The interviews were all transcribed verbatim, word for word. Two researchers (MV and SS) were deeply engaged with the data, methodically reevaluating the interviews and field notes. An inductive thematic analysis framework was employed to process the data and identify key themes. Thematic analysis serves as an interpretive strategy to uncover patterns of meaning within the transcribed interviews, providing insights into the phenomenon under study. This process included coding, categorization, and counting. Following the method described by Cohen et al. [16], the researchers independently immersed themselves in the data by reading and rereading the transcripts to grasp the overall data set. The data were organized and reorganized into higher-order themes until the research team reached a consensus. Attention then shifted to data interpretation, where the codes were analyzed to extract meaningful insights.

To ensure the credibility of the findings, the final organization of the themes and the content were collectively discussed and agreed upon by all research team members. The researchers compared the various extrapolated themes to establish dependability. Additionally, the initial themes were reviewed and either confirmed or revised by each participant during meetings involving all researchers.

The data analysis was carried out in Italian to avoid potential misunderstandings. The final results were later translated into English by professional translators during the editing of the article.

### 2.6. Ethical Considerations

For this study, the URC of the Azienda Ospedale-Università Padova’s ethical approval and the local hospital’s research protocol were provided with the following research number: 5766 of 18 May 2023.

### 2.7. Rigor and Reflexivity

To ensure transparency in this manuscript, the research utilized the Consolidated Criteria for Reporting Qualitative Research (COREQ) developed by Tong et al. [21]. A nurse specializing in qualitative research methodologies (MV) (female) was tasked with identifying eligible participants. For dependability, the analysis and results were scrutinized by two researchers (MV and SS). Throughout the research process, there was ongoing correspondence among team members to encourage reflexivity and thorough analysis. Various types of citations were employed to illustrate the connection between the data and the results.

## 3. Results

### 3.1. Characteristics of the Participants

The study sample (Table 1) comprised 18 caregivers, with 78% (n = 14) being female. Regarding the degree of kinship within the sample, it included ten daughters, four partners, two sisters, one mother, and one grandson. The average age of the caregivers in the sample was 57 years. The education level possessed by the caregivers was primarily at the high school level in most cases. In eight instances, the stroke survivors lived with the caregiver before the neurological event occurred.

Four themes emerged that are described in the following section:Sense of losing their own stroke survivors with these subthemes:
(a)Change in individual traits(b)Change in social relationshipsThe silent impact of confinement: life inside the homeRising to the challenge: caregiving supportLack of awareness about the needs during the transition

### 3.2. Sense of Losing Their Own Stroke Survivors

This theme emerged prominently in the major interviews. The perception of loss experienced by stroke survivors, as reported by their caregivers, was clearly evident in the majority of the interviews, revealing a complex and emotionally challenging situation. Caregivers underwent various stages of adjustment while navigating the alterations brought about by the stroke. Caregivers often encountered a profound shift in their relationship with the stroke survivor. The dynamics, at times, transitioned from a partnership to a caregiver–stroke-survivor dynamic, resulting in a perceived loss of the previous connection. This change extended to the caregiver’s social relationship with the external world, as their focus shifted toward the care of the stroke survivor. Social isolation adversely affected their emotional well-being, amplifying feelings of loneliness. Moreover, observing a loved one grapple with the physical, emotional, and cognitive challenges post-stroke was emotionally taxing for the caregivers. The loss of the person’s former self and the uncertainty surrounding the future contributed significantly to emotional distress. In MV02, the daughter expressed, “Once back home, I realized that it was no longer as it used to be…” and in MV11 son “She has changed; I see her differently. She forgets things; if I tell her something, she repeats it to me twice”. Additionally, in MV15, the sister expressed, “A completely different person, he has changed… [stroke] has changed him radically”.

In the first subtheme, caregivers delineated alterations to the character, characteristics, and role of stroke survivors. In interview MV03, the daughter described a shift in the mother’s mood, stating, “Also very moody, it was like that before as well, but now everything has become more pronounced”. The partner in MV05 indicated a change in the mood of the stroke survivor, remarking, “It’s not the same person as before anymore”. The interviews underscored a significant change in the role of stroke survivors. In MV06, the partner remarked, “Before, there was a man who helped with the things that men do. Now, I have to take care of it myself. I no longer have support, and on top of that, it’s not just about not peeing on himself, but washing him, changing him, and taking care of him is no small task”. Furthermore, the daughter in MV07 conveyed, “It all collapsed on me; everything became relative, and I found myself with a completely different husband [after the stroke]”.

The second subtheme revolves around changes in social relationships, with caregivers consistently expressing alterations to social dynamics for both themselves and the stroke survivors. Stroke, being a sudden and often life-altering event, can lead to profound physical, emotional, and social consequences for stroke survivors and their caregivers. Caregivers reported that stroke survivors often exhibited a reluctance to leave the house and avoided social engagements. In MV01, the partner mentioned, “…before, life was full and intense; now, clearly, we spend more time at home. Let’s skip the little walk for now, as he hasn’t started driving again. He still doesn’t feel as independent as before, let’s say”. Similarly, in MV05, the partner noted, “He never goes out, he doesn’t want to leave the house… It’s not the same as before, in short… he doesn’t want to leave the house”. Similarly, the partner (MV06) expressed, “Right now, we would need more to have a chat with old friends, or to see the world outside… but my husband doesn’t want to”. This shift in social relationships was attributed, in part, to the consequences of the stroke, particularly affecting speech. Communication difficulties were expressed in three interviews, where caregivers mentioned challenges in expressing themselves and understanding stroke survivors. In MV08, the caregiver remarked, “The situation is no longer like before. Dad used to talk a lot, but now he doesn’t say a word. It’s not that simple, but he is serene, and that’s what matters!”.

Within this subtheme, caregivers elucidated the relationship between the stroke survivor’s sense of freedom and their level of socialization. Sense of freedom is correlated with the level of autonomy and the ability to engage in activities similar to those performed before the acute event. In MV01, the partner expressed, “Because he’s not as free as he used to be, he used to go, he used to go, he used to do his shifts in S.C. (name of City)”. Similarly, in MV10, the sister revealed, “he has pain in his legs and is no longer free to go where he wants to get the newspaper, to do shopping alone”.

### 3.3. The Silent Impact of Confinement: Life inside the Home

Post-stroke events precipitate a metamorphosis in the personalities of stroke survivors, impacting their social interactions. Caregiver-perceived social support from peers and familiar acquaintances assumes a central role for both caregivers and stroke survivors in their convalescence and holistic welfare. It facilitates their management of stress, negotiation of caregiving duties, and preservation of health. As described by the daughter in MV07, “He does not leave the house as before but maintains relationships […] but at home, dad has many friends and thus still receives daily stimuli from people who come to see him”. Additionally, partners of stroke survivors in MV12 mentioned, “We don’t go out like we used to, but we manage to have a little contact with relatives, and this helps us a lot”. The significance of social and family support in adapting to the new situation emerged particularly in MV11: the son noted, “Now he has returned to being serene, not as happy as before, but he is doing better in the family environment”.

In such circumstances, voluntary associations play a vital role, offering a venue for gatherings. In MV06, the wife of a stroke survivor described it as “I feel… let’s say, abandoned, but fortunately, we have a good senior center here”.

### 3.4. Rising to the Challenge: Caregiving Support

Providing care for stroke survivors poses a significant challenge for their caregivers, requiring them to take on a diverse range of responsibilities crucial to the recovery and rehabilitation of the stroke survivor. Caregiving duties commence abruptly following the stroke event, intensifying the burden on caregivers who promptly assume the role of carer for their loved ones. This results in a heightened workload for caregivers, often leading to stress, burnout, and health issues arising from the inherent demands of their newfound responsibilities. The theme of burden became evident in MV02, making it particularly relevant. In this interview, the daughter of stroke survivors expressed, “I find it challenging to keep up with everything. I have a brother, but he lives 300 km away. So, you understand that there is a need for people who are physically close, not just emotionally. Therefore, this is really the major issue for me”. Moreover, this family was already burdened by other medical conditions. Similarly, in another interview (MV04), the daughter elaborated on the situation, stating, “On my own, I don’t know how I will be able to handle everything, especially since I also need to work due to our financial problems, and I can’t be available all day”. She further highlighted, “…has difficulty eating; the food needs to be prepared and then fed to him. It’s a very demanding task that, being alone, I have to get used to providing assistance every day, for everything […] Alone, he does nothing, and everything falls on me. I also have a family; I have a father who can’t help us, as he also has mobility difficulties”.

### 3.5. Lack of Awareness about the Needs during the Transition

The theme of lack of awareness about the needs during the transition, particularly in the realms of nutrition and daily care, was prominent in the interviews. It reflects the cultural sensitivity and approach to eating and treatment in Italy, which significantly influences the overall experience of care and healing. The average age of the caregivers in this interview was 57 years, and, for this demographic, food is not merely a physical necessity but often symbolizes security and stability. For individuals in this age group, food transcends mere consumption, embodying a sense of security and continuity. Additionally, food serves as a means of maintaining social bonds and foster a sense of community. Consequently, food is not perceived solely as an act of consumption, but rather as a symbol of security, health, and communal belonging.

In one interview (MV11), a caregiver initially expressed a sense of normalcy in their approach to dietary habits, stating, “He follows a regular diet, with minced meat instead of whole, just to facilitate”. However, he later added, “We realized that there is a significant risk in eating and drinking because, instead of swallowing through the esophagus, he aspirates it, and he has experienced bronchitis twice”. Similarly, in another interview (MV12), the partner described the situation, saying, “Some foods go the wrong way, but, in short, we consider the matter quite controllable. There is a bit of difficulty in swallowing, but it is not severe”. The theme of daily care emerged in the interview MV15. Here, the caregiver discussed the initial difficulty in bathing her sister where: “I couldn’t do it initially… I learned to manage it when she came home”. The daughter in MV16 expressed “The injuries resulting from us raising her from the bed and then dressing her slowly improved […] for walking and all the other needs we were able to manage alone”.

## 4. Discussion

The purpose of this study was to explore the lived experiences of caregivers of older stroke survivors regarding expectations in the care pathway during the transitional home program. The predominantly female sample consisted of ten children, four partners, two sisters, one mother, and one grandson, providing a diverse representation of kinship within the caregiving role.

The main results revealed a profound sense of loss experienced by older stroke survivors, as described by caregivers. Caregivers witnessed significant changes in individual traits, including mood and personality alterations, along with significant transformations in the social roles of stroke survivors, leading to a perceived loss of their previous identity and roles. This loss extended beyond individual traits to encompass changes in social relationships, as stroke survivors often experienced a reluctance to engage in social activities, leading to feelings of social isolation for both caregivers and survivors. These findings are consistent with previous research highlighting the emotional toll of stroke on both survivors and caregivers, as well as the challenges of adjusting to post-stroke life [8,15,22,23]. Since a stroke represents an acute event that abruptly alters the autonomy of survivors, it necessitates rapid adaptation by both survivors and caregivers to the new circumstances.

In the second theme, caregivers emphasized social relationships by focusing on their perception of the social relationships of stroke survivors, both in terms of providing emotional support and in terms of facilitating the adaptation process for stroke survivors and their caregivers. They described these relationships as a fundamental element, given that the adaptation and adjustment process is highly subjective and delicate [24]. Caregivers highlighted the significance of maintaining social connections and receiving support from peers and family members when coping with the challenges of caregiving and stroke survival. This highlights the vital role of social support, with voluntary associations playing a significant part in helping caregivers adapt to the new reality. This finding aligns with previous research emphasizing the importance of social support in enhancing the well-being of caregivers and stroke survivors during the transition to the home environment [11,14].

Caregivers shared the daily challenges they faced when managing the physical and emotional needs of stroke survivors. They described feeling overwhelmed by the demands of caregiving, particularly in the absence of adequate support systems. This increased workload and stress experienced by caregivers as they assume the responsibilities of caring for stroke survivors underscore the need for targeted interventions and support services to alleviate the burden on caregivers and prevent burnout. The theme focusing on the lack of awareness about the needs during the transition highlights some caregivers’ lack of awareness of certain needs, such as nutritional issues or daily care, leading to unexpected challenges. Caregivers described grappling with issues such as difficulty swallowing and managing daily tasks such as bathing and dressing, underscoring the need for tailored support and resources to address these challenges. This finding underscores the importance of comprehensive care planning and ongoing support for caregivers to ensure the well-being of both caregivers and stroke survivors during the transition to the home environment.

This study has several implications. Firstly, it provides an in-depth overview of the challenges faced by caregivers of stroke survivors one month after discharge from the hospital. Recognizing and addressing the sense of loss experienced by both stroke survivors and their caregivers is essential for fostering a supportive and understanding environment during the recovery process. Social support plays an integral role in the overall care of stroke survivors, contributing not only to physical recovery, but also to emotional and social well-being, thereby promoting a more comprehensive and effective rehabilitation process. These social factors act as moderators in the quality of life of stroke survivors; in fact, the lack of social support is associated with poorer physical function, increased depression, and reduced vitality in stroke survivors [10].

Secondly, from the interviews, the theme of unrecognized needs emerged, a topic not yet extensively described in the literature. Caregivers of stroke survivors are often unaware of the risks these individuals may face, particularly concerning dietary issues and personal care, as highlighted in the interviews. A cross-sectional study on patients with severe dementia and dysphagia found that caregiving often demands meticulous attention and specialized care, placing a significant burden on caregivers [25]. It is from these unrecognized needs that healthcare professionals can develop tailored interventions to support caregivers and improve the transition from the hospital to the home environment for stroke survivors and their caregivers [26].

The implications of this study highlight the need for targeted interventions to address social transformation, increased workload, and unrecognized needs among the caregivers of stroke survivors. Specifically, the findings underscore the importance of developing strategies to enhance social support networks, provide practical resources to reduce the caregiving burden, and increase awareness of the potential challenges that caregivers face. Future research should focus on identifying and implementing these strategies, with the goal of creating tailored programs that directly address the unique challenges highlighted in this study. Moreover, a deeper understanding of these dynamics is essential for implementing healthcare programs that can improve the quality of life for both caregivers and stroke survivors. Such programs could include caregiver education initiatives, integration of social support services within the care pathway, and provision of respite care to alleviate the emotional and physical impact on caregivers. By broadening the scope of interventions, it is possible to develop more comprehensive care models that not only focus on the physical recovery of stroke survivors, but also address the well-being of those who support them.

### Strengths and Limitations

This study was exclusively conducted within a specific Italian region, Veneto, and in a single hospital. This approach introduces potential sources of bias due to subtle cultural differences between countries. The use of video calls for interviews imposed limitations on recording detailed field notes, particularly regarding environmental descriptions. Another potential limitation stems from the predominantly female composition of the sample, which may introduce a gender-related bias in the perspective of the phenomenon. Additionally, the use of a phenomenological approach, while valuable for capturing lived experiences, may not fully encompass the broader contextual factors influencing caregiving. Future research should aim to include more diverse samples and employ mixed-method approaches to enrich the depth of our understanding.

## 5. Conclusions

This qualitative study has delved into the multifaceted experiences of caregivers during the transitional care period for stroke survivors, providing valuable insights into the challenges they encounter and the expectations they hold. The findings highlight the profound impact of stroke on caregivers, affecting both individual traits and social relationships of the stroke survivors and their caregivers, emphasizing the need for a nuanced understanding of caregivers’ roles and responsibilities.

Recognizing the pivotal role of caregivers is crucial, and tailoring interventions to meet their unique needs becomes an essential step toward enhancing the overall well-being of both caregivers and stroke survivors.

## Figures and Tables

**Table 1 ijerph-21-01276-t001:** Socio-demographic characteristics of the caregivers (N = 18).

ID	Age	Sex	Status	Type of Work	Hour Work/Week	Educational Status	Relationship with Stroke Survivor	Children	N (Children)	Number of Family Members	Living with a Relative before the Stroke	Barthel Index of Stroke Survivor
Mv01	78	F	Married	Retired	-	High school	Partner	Yes	4	2	Yes	85
Mv02	51	F	Married	Other (housewife)	-	High school	Daughter	Yes	2	5	No	60
Mv03	55	F	Single	Other (educator)	-	High school	Daughter	No	-	1	No	70
Mv04	57	F	Married	Worker	40	High school	Daughter	Yes	1	3	No	55
Mv05	75	F	Married	Retired	-	Middle school	Partner	No	-	2	Yes	65
Mv06	71	F	Married	Retired	-	Middle school	Partner	Yes	1	2	Yes	80
Mv07	41	F	Single	Other (health care assistant)	36	High school	Daughter	No	-	1	No	75
Mv08	55	F	Married	*Worker*	40	Degree	Daughter	Yes	1	3	No	90
Mv09	51	M	Married	*Worker*	36	Middle school	Son	Yes	2	4	No	75
Mv10	76	F	Married	Retired	-	Primary school	Sister	Yes	3	2	No	45
Mv11	50	M	Married	*Worker*	40	High school	Son	No	-	3	Yes	60
Mv12	74	F	Married	Retired	-	High school	Partner	Yes	3	2	Yes	85
Mv13	32	F	Single	*Worker*	40	High school	Grandson	No	-	4	No	65
Mv14	47	F	Separated/divorced	*Worker*	36	Degree	Daughter	Yes	2	4	No	75
Mv15	52	M	Single	*Worker*	36	Middle school	Sister	No	-	2	Yes	40
Mv16	44	M	Single	Other (health care assistant)	36	Degree	Son	No	-	2	Yes	65
Mv17	56	F	Married	Other (housewife)	-	Middle school	Daughter	Yes	4	6	No	70
Mv18	68	F	Separated/divorced	Retired	-	High school	Mother	Yes	1	2	Yes	85

## Data Availability

Additional data are available from the corresponding author upon request.

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
