# Peer review of "The Lived Experience of Caregivers in the Older Stroke Survivors’ Care Pathway during the Transitional Home Program—A Qualitative Study"

_ijerph, 2024, doi:10.3390/ijerph21101276_

Round 1
Reviewer 1 Report
Comments and Suggestions for Authors
1. Please see detailed comments in the attached file.
2. The manuscript has some very dated references. It is recommended that more recent work is cited - even if they do not relate directly to the care of individuals who have had a stroke, but address various aspect of caregiving. A suggestion is to read Sharma et al. (2024) Caregiver experiences with dementia-related feeding/eating difficulties.
3. It would have been beneficial to report the number of disabilities the individual has in relation to the burden of caregiving. It has been shown that the more disabilities faced, the greater the challenge of caregiving. Additionally, the greater the disability and dependence on the caregiver, the greater the emotional and psychological stress. None of these aspects have been addressed in this manuscript signaling a potential flaw in data collection/reporting of data.

Comments on the Quality of English LanguageIt is suggested that the Authors use a more academic style of writing throughout the manuscript.
Author Response
1.Please see detailed comments in the attached file.
Thank you for the time you spent for reviewing this article.
- The manuscript has some very dated references. It is recommended that more recent work is cited - even if they do not relate directly to the care of individuals who have had a stroke but address various aspect of caregiving. A suggestion is to read Sharma et al. (2024) Caregiver experiences with dementia-related feeding/eating difficulties.
Thank you for your suggestions. We changed several references with more recent references. We read and appreciate the study that you suggest. We cited this study in discussion.
- It would have been beneficial to report the number of disabilities the individual has in relation to the burden of caregiving. It has been shown that the more disabilities faced, the greater the challenge of caregiving. Additionally, the greater the disability and dependence on the caregiver, the greater the emotional and psychological stress. None of these aspects have been addressed in this manuscript signaling a potential flaw in data collection/reporting of data.
Thank you for your comment. We appreciate your suggestion regarding the relationship between the number of disabilities and the burden of caregiving. We have included the Barthel Index scores of stroke survivors in Table 1, which provides an indication of their level of disability and dependence on caregivers. We understand that the number and severity of disabilities can significantly impact the caregiving burden, and we will consider expanding the analysis in future studies to better capture this dimension.
Reviewer 2 Report
Comments and Suggestions for Authors
The lived experience of caregivers in the older stroke survivors’ care pathway during the transitional home program - A qualitative study
This important study aimed to explore the experiences of caregivers of older stroke survivors regarding their expectations of the care pathway during the Transitional Home Program. In my view, as disability among stroke survivors becomes a growing phenomenon in our aging society (along with related chronic diseases), family caregivers play a key role in the transition to home. In this context, these caregivers often feel unprepared to fulfill this responsibility, which may result in increased stress and distress. Given that this study employed a qualitative design and a phenomenological approach, the current paper has the potential to make a valuable contribution to the literature on informal caregivers in the context of stroke. Below, I offer some suggestions that may be helpful in revising the paper. I encourage the authors to consider the following suggestions as they continue to develop the manuscript.
General comments:
1) A more comprehensive literature review would undoubtedly prove beneficial for the development of this article, in the case that the words limit allows for it.
2) What are the theoretical foundations or framework of this article? Does the article present or otherwise address specific types of Informal Caregiver Model?
Abstract:
3) The abstract is succinct and provides a clear overview of the study's objective. To enhance the rationale, it would be beneficial to include a statement indicating that this phenomenon has not been extensively studied in the present context, particularly through the lens of qualitative methodology.
Introduction:
The introduction is well-written and effectively delineates the areas where there are gaps in the existing literature. Nevertheless, this section is incomplete in several respects and thus requires further elaboration.
4) The article's unique contribution is not adequately addressed. It would be beneficial to include a detailed account of the distinctive value of the data, which is based on a comprehensive examination of the personal experiences of these caregivers. Additionally, it would be advantageous to present insights and recommendations in terms of their practical implications. For instance, the following sentence could be expanded:
"The aim of this qualitative study is to explore the lived experiences of caregivers of older stroke survivors regarding their expectations in the care pathway during the transitional home program".
5) It is necessary to provide a rationale for the qualitative study design and justify the in-depth data collection. Specifically, to provide an explanation as to whether the specific condition of the phenomenon justifies the use of qualitative inquiry of caregivers of older stroke survivors and their expectations of the care pathway during the Transitional Home Program.
It may be preferable to categorize this paper as a case study design, as this approach is particularly effective for elucidating the experiences of caregivers of older stroke survivors within the context of the Transitional Home Program. Also, the study includes an in-depth overview of the challenges faced by caregivers of stroke survivors one month after discharge at home. However, this modification should only be implemented only if it is applicable and can be integrated into the current methodology.
6) What is the principal research question? Additionally, this is also applicable to qualitative studies.
Materials and Methods:
7) Study design is well described as well as the setting and inclusion and/or exclusion criteria.
8) Data Analysis - A more detailed account of the thematic analysis process would be beneficial.
9) As stated in the Abstract, eighteen caregivers of stroke survivors participated in individual, digitally recorded, semi-structured interviews. However, the information regarding the research interview guidelines is absent.
o In what manner is an initial guided interview designed? Is it done by combining questions derived from existing literature with additional open-ended questions?
Discussion:
10) In accordance with comment no. 2 above, it is essential to identify, develop, and discuss both a research question and a theoretical foundation in the discussion section. In particular, this section should provide an exposition of the theoretical contributions of the study.
11) The implications are introduced in a cursory and generic manner. In alignment with my initial remarks in the 'Introduction' section, this point merits further elaboration.
To conclude, in light of the qualitative design employed in this study and the significance of this subject matter, the present paper has the potential to make an incremental contribution to the existing literature on informal caregivers in the context of stroke. I encourage the authors to consider enhancing their article and resubmit a revised paper.
Author Response
Reviewer 2
This important study aimed to explore the experiences of caregivers of older stroke survivors regarding their expectations of the care pathway during the Transitional Home Program. In my view, as disability among stroke survivors becomes a growing phenomenon in our aging society (along with related chronic diseases), family caregivers play a key role in the transition to home. In this context, these caregivers often feel unprepared to fulfill this responsibility, which may result in increased stress and distress. Given that this study employed a qualitative design and a phenomenological approach, the current paper has the potential to make a valuable contribution to the literature on informal caregivers in the context of stroke. Below, I offer some suggestions that may be helpful in revising the paper. I encourage the authors to consider the following suggestions as they continue to develop the manuscript.
Thank you for your thoughtful comments.
General comments:
1) A more comprehensive literature review would undoubtedly prove beneficial for the development of this article, in the case that the words limit allows for it.
Thank you for suggestions. In this new version of the manuscript, we implemented as you suggested.
2) What are the theoretical foundations or framework of this article? Does the article present or otherwise address specific types of Informal Caregiver Model?
Thank you for suggestions. In this new version of the manuscript, we implemented as you suggested in Introduction section.
Abstract:
3) The abstract is succinct and provides a clear overview of the study's objective. To enhance the rationale, it would be beneficial to include a statement indicating that this phenomenon has not been extensively studied in the present context, particularly through the lens of qualitative methodology.
Thank you for your suggestion. We modified it.
Introduction:
The introduction is well-written and effectively delineates the areas where there are gaps in the existing literature. Nevertheless, this section is incomplete in several respects and thus requires further elaboration.
We are sorry for this. We integrated several aspects in the introduction. We hope that this could be exhaustive
4) The article's unique contribution is not adequately addressed. It would be beneficial to include a detailed account of the distinctive value of the data, which is based on a comprehensive examination of the personal experiences of these caregivers. Additionally, it would be advantageous to present insights and recommendations in terms of their practical implications. For instance, the following sentence could be expanded:
"The aim of this qualitative study is to explore the lived experiences of caregivers of older stroke survivors regarding their expectations in the care pathway during the transitional home program".
Thank you for your suggestion. We modified it in page 2, lines 77 – 85.
5) It is necessary to provide a rationale for the qualitative study design and justify the in-depth data collection. Specifically, to provide an explanation as to whether the specific condition of the phenomenon justifies the use of qualitative inquiry of caregivers of older stroke survivors and their expectations of the care pathway during the Transitional Home Program.
It may be preferable to categorize this paper as a case study design, as this approach is particularly effective for elucidating the experiences of caregivers of older stroke survivors within the context of the Transitional Home Program. Also, the study includes an in-depth overview of the challenges faced by caregivers of stroke survivors one month after discharge at home. However, this modification should only be implemented only if it is applicable and can be integrated into the current methodology.
Thank you for your suggestion. We have reviewed and clarified the relevant section. We believe our study should be classified as a qualitative study rather than a case study design, as we did not focus on one or a few specific cases. Instead, we conducted an in-depth exploration of the experiences of a broader group of caregivers of older stroke survivors, achieving data saturation to capture the various nuances and complexities of their experiences.
6) What is the principal research question? Additionally, this is also applicable to qualitative studies.
Thank you for suggestions. In this new version of the manuscript, we implemented as you suggested. Specifically, we added in Introduction section, page 2, lines 83-85.
Materials and Methods:
7) Study design is well described as well as the setting and inclusion and/or exclusion criteria.
Thank you!!
8) Data Analysis - A more detailed account of the thematic analysis process would be beneficial.
Thank you for suggestions. In this new version of the manuscript, we implemented as you suggested in the “Data Analysis”
9) As stated in the Abstract, eighteen caregivers of stroke survivors participated in individual, digitally recorded, semi-structured interviews. However, the information regarding the research interview guidelines is absent.
In what manner is an initial guided interview designed? Is it done by combining questions derived from existing literature with additional open-ended questions?
In this new version of the manuscript, we implemented as you suggested in “Data Collection”.
Discussion:
10) In accordance with comment no. 2 above, it is essential to identify, develop, and discuss both a research question and a theoretical foundation in the discussion section. In particular, this section should provide an exposition of the theoretical contributions of the study.
Thank you for suggestions. In this new version of the manuscript, we implemented as you suggested.
11) The implications are introduced in a cursory and generic manner. In alignment with my initial remarks in the 'Introduction' section, this point merits further elaboration.
Thank you for suggestions. In this new version of the manuscript, we implemented as you suggested.
To conclude, in light of the qualitative design employed in this study and the significance of this subject matter, the present paper has the potential to make an incremental contribution to the existing literature on informal caregivers in the context of stroke. I encourage the authors to consider enhancing their article and resubmit a revised paper.
Thank you for your constructive feedback and encouragement. We will resubmit the revised paper, incorporating your valuable feedback to strengthen our contribution to the field.
Round 2
Reviewer 2 Report
Comments and Suggestions for Authors
In their response to the comments and by adding explanations, the authors provide a satisfactory improvement to the article.